# Impact of restriction of over-the-counter sales of antimicrobials on antimicrobial resistance in *Escherichia coli* from community-onset urinary tract infections in inner São Paulo State, Brazil

**Fernanda Saad Rodrigues, Helena Ribeiro Aiello Amat, Carlos Magno Castelo Branco Fortaleza** [ID] *

Department of Infectious Diseases, Botucatu School of Medicine, São Paulo State University (UNESP), City of Botucatu, São Paulo State, Brazil

* carlos.fortaleza@unesp.br

**Data Availability Statement:** Study database is included as a supplementary file.

## Abstract

### Background

Antimicrobial resistance in community-associated infections is an increasing worldwide concern. In low-to-middle income countries, over-the-counter (OTC) sales of antimicrobials without medical prescription have been blamed for increasing consumption and resistance. We studied the impact of restriction of OTC sales of antimicrobials in Brazil (instituted in October 2010) on resistance trends of *Escherichia coli* from community-onset urinary tract infections.

### Methods

We analyzed monthly resistance trend of Escherichia coli from community-onset urinary tract infections from 2005 through 2018. The data were submitted to interrupted time series analysis in both linear and Poisson regression models.

### Results

We found impact on cefazolin (p<0.001) and amikacin (p<0.001) resistance as immediate impact of the intervention, and no beneficial impact on resistance to ciprofloxacin, ceftriaxone or sulfamethoxazole-trimethoprim.

### Conclusion

At the present study, we found that OTC sales restriction did not generally impact on antimicrobial resistance.

**Funding:** HRAA received a student research grant from São Paulo State Research Foundation (FAPESP, grant #2018/17210-1), with CMCBF as her advisor. The funding agency did not played no role in study design, methodology, nor in the writing of the manuscript.

**Competing interests:** The authors have declared that no competing interests exist.

**Abbreviations:** COUTI, community-onset urinary tract infections; ITS, interrupted time series (analysis); SMX-TMP, sulfamethoxazole-trimethoprim.

## Background

In the past decade, the World Health Organization (WHO) recognized antimicrobial resistance as a major threat for public health worldwide. Countries were urged to develop their own action plans to prevent and control the emergence and spread of antimicrobial-resistant pathogens [1, 2]. This is a major challenge for low-to-middle income countries, which have experienced recent increases in consumption of antimicrobials [3]. In those countries, over-the-counter (OTC) sales account for a relevant proportion of excessive use of antimicrobial agents [4, 5]. Laws restricting or prohibiting OTC sales of antimicrobials have been issued, but their impact is currently not straightforward [6, 7].

The population drivers of antimicrobial resistance are complex. A vast array of direct and herd effects interplay in the continuous process of emergence and spread of resistant bacterial strains [8]. Even though secular trends demonstrate clear association between the introduction of antimicrobials in clinical practice and the emergence and spread of resistance phenotypes, this association is not straightforward [9]. Therefore, the impact of regulations must be assessed not only regarding antimicrobial sales, but also focusing on their outcomes of interest, i.e., on the time trends of resistance among common pathogens.

A ban on non-prescription sales of antimicrobials in private drugstores was issued by Brazilian Sanitary Agency (ANVISA) in October 2010. The impact of this legal measure was uneven across the country. In fact, there was a significant reduction in sales of antimicrobials in pharmacies in the Southern and Southeastern states, which have a higher human development index (as compared to Northern, Northeastern and Midwest, and corresponds to the study zone. The effect was small in the less developed areas [10, 11]. Most importantly, the impact on antimicrobial resistance is uncertain in Brazil and in our region.

With that in mind, we use time series analysis to investigate the impact of ANVISA regulation on antimicrobial resistance in *Escherichia coli* isolated in urine from patients with community-onset urinary tract infection (COUTI) in inner São Paulo State, Brazil.

## Methods

### Ethical approval

This study was approved by the Committee for Ethics in Human Research in Botucatu Medical School (register: CAAE 30000014.1.0000.5411), city of Botucatu, São Paulo State, Brazil. The ethics committee waived the requirement for informed consent. Data from all patients was initially checked for duplications, and subsequently anonymized before final validation and statistical analyses.

### Study setting and design

We conducted a natural experiment study based on ecological data on incidence and antimicrobial resistance among *E. coli* isolates from COUTI in patients living in the Botucatu Health Administrative Area (BHAA), inner São Paulo State, Brazil. That area comprises 13 municipalities located around Botucatu city (22°53′25″S, 48°27′19″W), with a total of 300,000 inhabitants. Microbiological tests from public primary care units and outpatient services are performed in the laboratory from Botucatu Medical School teaching hospital.

### Operational procedures

We collected data on microbiological results from COUTI in the period from January 2005 through December 2018. Those cultures were obtained from patients with suspected COUTI, cared for in public primary care and outpatient facilities in the BHAA. We recorded resistance

of *E. coli* isolates to several antimicrobials, mainly the most used for the treatment of COUTI in our country: cefazolin, ceftriaxone, ciprofloxacin, amikacin and sulfamethoxazole-trimethoprim (SMX-TMP). We chose to address those antibiotics based on their use in outpatients or inpatients therapy of COUTI. Antimicrobial resistant tests were performed in Microscan Autoscan system (Beckman Coulter, Pasadena, CA) and followed standards from the Clinical Laboratory Standards Institute (CSLI) [12]. Duplicate results (defined as those obtained within a 30-day period) and those in which a nosocomial origin was proved were excluded from the study.

### Time series analysis

Time series of monthly incidence of pathogens and resistance patterns were submitted to interrupted time series analysis (ITS) through segmented regression in the *itsa* package of STATA 14 (Statacorp, College Station, TX), as previously described [13]. Briefly, we analyzed the impact of the OTC sales ban in October 2010 on the immediate and late trends of antimicrobial resistance. We complemented this analysis with Poisson Regression models, adjusted for time trends and seasons.

In order to further investigate seasonality, Box-Jenkins models were fit to time series of incidence and antimicrobial resistance. We used the following parameters for models: regular (1,0,1), seasonal (1,0,1), with seasonality in 12 months [14]. Those analyses were performed in NCSS 9 (LLC, Kaysville, UT).

## Results

In the study period, there were 40,814 positive cultures from COUTI patients, of which 31,162 (76.4%) recovered *E. coli*. The cumulative incidence was 10,686 per 100,000 inhabitants. There was a continuous trend towards increase in the isolates (incidence rate ratio [IRR] for month, 1.004; 95% confidence interval, 1.003–1.005; $P<0.001$), with peaks in summer (IRR, 1.063; 95% 1.031–1.097; $P<0.001$, as compared to winter). Those findings imply an average increasing trend in rates of 0.4% per month, as well as an overall increase of 6.3% during summer. While the overall increase may be attributable to greater access to diagnostic tests, the seasonal findings are consistent and were corrected excluding long term trends. Coherently, we found significant coefficients ($P<0.001$) for autoregressive, moving average and–most importantly—seasonal parameters in Box Jenkins models.

Results from ITS analysis of resistance trends are presented in **Fig 1** and **Table 1** presents complementary season-adjusted Poisson regression models. Overall, immediate decrease in resistance was detected in all models only for cefazolin and amikacin. No beneficial impact on long-term trends was identified for the resistance to antimicrobials of interest.

In an opposite direction, there was immediate increase in resistance to ceftriaxone (ITS and Poisson models) and ciprofloxacin (Poisson regression), and an upward change in long-term trends of resistance to TMP-SMX (ITS and Poisson regression) and cefazolin (Poisson regression). The impact of seasons in Poisson multivariable models varied, and no resistance rate presented significant seasonal parameters in Box-Jenkins models.

## Discussion

COUTI are unique among bacterial infections, because they are by far the ones for which cultures are most frequently collected. Therefore, they are expected to provide a reliable proxy of antimicrobial resistance trends in the community setting [15]. Our study aimed to analyze the impact of restriction of OTC sales of antimicrobials based on trends in resistance in *E. coli*, the most frequent uropathogen.

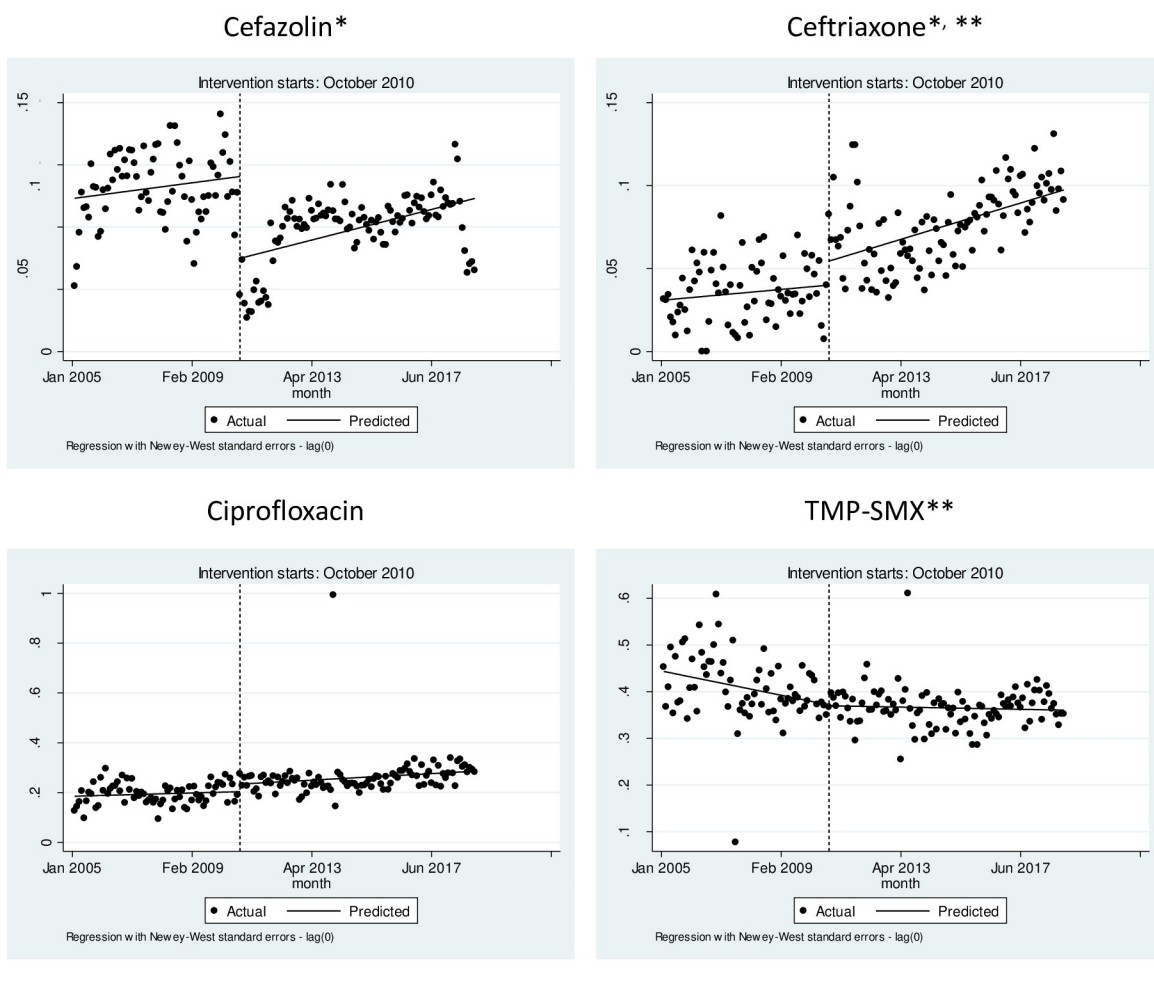

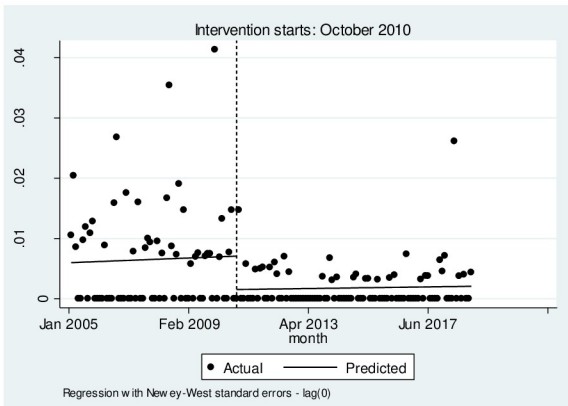

**Fig 1. Interrupted time series (ITS) analysis (linear regression models) of time trends of antimicrobial resistance (expressed in Y axis in proportion) in *Escherichia coli* recovered from urine cultures of patients with community-onset urinary tract infection in inner São Paulo State, Brazil, before and after the prohibition of over-the-counter sales of antimicrobials (antimicrobial sales without medical prescription).** TMP-SMX, Sulfamethoxazole-trimethoprim. Notice that the number of months with zero resistance to Amikacin limits the accuracy of ITS. *Immediate impact of the intervention ($P<0.05$). ** Impact of intervention on time trends ($P<0.05$).

**Table 1. Poisson regression analysis (adjusted for seasonal variation) of interrupted time series of resistance trends in *Escherichia coli* recovered from urine cultures of patients with community-onset urinary tract infection in inner São Paulo State, Brazil, before and after the prohibition of over-the-counter sales of antimicrobials (antimicrobial sales without medical prescription).**

| Antimicrobial resistance / predictors | IRR (95%CI) | P |
|---|---|---|
| ***Cefazolin*** | | |
| Time (months) | **1.001 (1.000–1.003)** | **0.03** |
| Intervention* | **0.588 (0.545–0.634)** | **<0.001** |
| Intervention_time** | **1.002 (1.001–1.004)** | **0.007** |
| Season | | |
| *Winter (reference)* | . . . | . . . |
| *Spring* | 0.985 (0.938–1.034) | 0.55 |
| *Summer* | 0.999 (0.953–1.047) | 0.98 |
| *Autumn* | **1.059 (1.011–1.111)** | **0.02** |
| ***Ceftriaxone*** | | |
| Time (months) | 1.004 (0.998–1.010) | 0.23 |
| Intervention* | **1.334 (1.041–1.709)** | **0.02** |
| Intervention_time** | 1.003 (0.997–1.009) | 0.39 |
| Season | | |
| *Winter (reference)* | . . . | . . . |
| *Spring* | 0.932 (0.825–1.052) | 0.26 |
| *Summer* | 1.010(0.898–1.137) | 0.86 |
| *Autumn* | 0.925 (0.820–1.045) | 0.21 |
| ***Cipro*** | | |
| Time (months) | 1.002 (0.999–1.005) | 0.10 |
| Intervention* | **1.137 (1.017–1.271)** | **0.02** |
| Intervention_time** | 0.999 (0.997–1.002) | 0.88 |
| Season | | |
| *Winter (reference)* | . . . | . . . |
| *Spring* | **0.931 (0.873–0.994)** | **0.03** |
| *Summer* | **1.074 (1.010–1.142)** | **0.02** |
| *Autumn* | 0.964 (0.904–1.027) | 0.26 |
| **SMX-TMP** | | |
| Time (months) | **0.997 (0.995–0.999)** | **0.01** |
| Intervention* | 0.971 (0.894–1.056) | 0.50 |
| Intervention_time** | **1.002 (1.000–1.004)** | **0.03** |
| Season | | |
| *Winter (reference)* | . . . | . . . |
| *Spring* | 0.976 (0.926–1.028) | 0.37 |
| *Summer* | 1.005 (0.955–1.057) | 0.85 |
| *Autumn* | 1.002 (0.951–1.054) | 0.95 |
| ***Amikacin*** | | |
| Time (months) | 1.003 (0.988–1.017) | 0.35 |
| Intervention* | **0.181 (0.075–0.437)** | **<0.001** |
| Intervention_time** | 1.003 (0.985–1.021) | 0.38 |
| Season | | |
| *Winter (reference)* | . . . | . . . |
| *Spring* | 0.969 (0.518–1.810) | 0.92 |
| *Summer* | 1.131 (0.636–2.012) | 0.67 |

(*Continued*)

**Table 1.** (Continued)

| Antimicrobial resistance / predictors | IRR (95%CI) | *P* |
|---|:---:|:---:|
| *Autumn* | 1.272 (0.726–2.222) | 0.40 |

**Note.** Significant (*P*<0.05) results are presented in boldface. IRR, Incidence Rate Ratio; CI, Confidence Interval; SMX-TMP, sulfamethoxazole-trimethoprim.

*Immediate impact of the intervention.

** Impact of intervention on time trends.

Even though the effect of the restriction on sales of antimicrobials in São Paulo state was relevant [11, 12], we found little to no impact on antimicrobial resistance in COUTI *E. coli*. Trends of resistance to ciprofloxacin were not impacted. Resistance to ceftriaxone accelerated after 2010, while the decreasing trend in TMP-SMX resistance slowed. Immediate or long-term changes after the sales restriction was noticed only for first generation cephalosporins and aminoglycosides, which are not "first line" therapeutic choices for COUTI [16].

Disappointing as our findings may be, they are evidently not an argument against restricting OTC sales of antimicrobials. Instead, they point out that restriction was not sufficient for stopping increasing trends in resistance to widely used therapeutic agents. Presumably, there are other drivers of the emergence and spread of resistance. Regarding *E. coli* and other Enterobacteriaceae, the fecal carriage of resistant strains in community settings [17], arising from food chain and eventually spread by international travel [18] may be among those drivers. One health approaches have demonstrated the importance of antimicrobial use in agriculture and the dispersion of those drugs in the environment in the causal chain of resistance. Policies that do not address those drivers from a wide ecological perspective are therefore likely to fail [19].

Our study was limited by its small geographic range and relatively short-term analysis. Also, it addressed only one pathogen from a single clinical syndrome, albeit the most frequently identified in the community setting. Finally, it did not include in its analysis the antimicrobial resistance arising and spreading within healthcare-settings. In those settings, other aspects such as hospital connectiveness and patient sharing may play a role in the dissemination of resistance phenotypes [20]. However, there are also strengths in our study. Antimicrobial resistance rates evolve and will obviously not change from one day to another, so our long-term analysis seemed appropriate. We analyzed results from more than 30,000 nonduplicate urine cultures collected over 14 years, using the appropriate ITS approach for natural experiments [13].

## Conclusion

In conclusion, we found that legislation of OTC sales of antimicrobials in drugstores did not present a relevant impact on the resistance of *E. coli* to antimicrobials widely used in clinical practice. Therefore, wider interventions should be consider for prevention and control of antimicrobial resistance in community settings.

## Supporting information

**S1 File.**
(XLSX)

## Acknowledgments

This study is part of PhD thesis of FSR, with CMCBF as her advisor.

## Author Contributions

**Conceptualization:** Fernanda Saad Rodrigues, Carlos Magno Castelo Branco Fortaleza.

**Data curation:** Fernanda Saad Rodrigues, Helena Ribeiro Aiello Amat, Carlos Magno Castelo Branco Fortaleza.

**Formal analysis:** Fernanda Saad Rodrigues, Helena Ribeiro Aiello Amat, Carlos Magno Castelo Branco Fortaleza.

**Funding acquisition:** Helena Ribeiro Aiello Amat, Carlos Magno Castelo Branco Fortaleza.

**Investigation:** Fernanda Saad Rodrigues, Carlos Magno Castelo Branco Fortaleza.

**Methodology:** Carlos Magno Castelo Branco Fortaleza.

**Project administration:** Carlos Magno Castelo Branco Fortaleza.

**Supervision:** Carlos Magno Castelo Branco Fortaleza.

**Validation:** Fernanda Saad Rodrigues, Helena Ribeiro Aiello Amat, Carlos Magno Castelo Branco Fortaleza.

**Visualization:** Fernanda Saad Rodrigues, Helena Ribeiro Aiello Amat, Carlos Magno Castelo Branco Fortaleza.

**Writing – original draft:** Fernanda Saad Rodrigues, Helena Ribeiro Aiello Amat, Carlos Magno Castelo Branco Fortaleza.

**Writing – review & editing:** Fernanda Saad Rodrigues, Helena Ribeiro Aiello Amat, Carlos Magno Castelo Branco Fortaleza.

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
