## [Decision Letter · Decision Letter 0]

18 Mar 2021

PONE-D-21-00235

Impact of restriction of over-the-counter sales of antimicrobials on antimicrobial resistance in Escherichia coli from community-onset urinary tract infections in inner São Paulo State, Brazil.

PLOS ONE

Dear Dr. Fortaleza,

Thank you for submitting your manuscript to PLOS ONE. After careful consideration, we feel that it has merit but does not fully meet PLOS ONE’s publication criteria as it currently stands. Therefore, we invite you to submit a revised version of the manuscript that addresses the points raised during the review process.

Please proceed to revise the manuscript paying special attention to the comments provided by reviewer number 2. We look forward to your revised version.

We look forward to receiving your revised manuscript.

Kind regards,

Monica Cartelle Gestal, PhD

Academic Editor

PLOS ONE

Journal Requirements:

2. In ethics statement in the manuscript and in the online submission form, please provide additional information about the patient records/samples used in your retrospective study. Specifically, please ensure that you have discussed whether all data/samples were fully anonymized before you accessed them and/or whether the IRB or ethics committee waived the requirement for informed consent. If patients provided informed written consent to have data/samples from their medical records used in research, please include this information.

Reviewers' comments:

Reviewer's Responses to Questions

**Comments to the Author**

1. Is the manuscript technically sound, and do the data support the conclusions?

Reviewer #1: Yes

Reviewer #2: Partly

2. Has the statistical analysis been performed appropriately and rigorously? 

Reviewer #1: Yes

Reviewer #2: I Don't Know

3. Have the authors made all data underlying the findings in their manuscript fully available?

Reviewer #1: Yes

Reviewer #2: Yes

4. Is the manuscript presented in an intelligible fashion and written in standard English?

Reviewer #1: Yes

Reviewer #2: Yes

5. Review Comments to the Author

Reviewer #1: I found this manuscript very well written, in a clear and concise manner.

All the findings are very well presented and the analysis of data is well done.

The findings are mostly negative which is very interesting considering how much emphasis is put on controlling antimicrobial drug use as a way to control resistance.

I feel that the manuscript should be accepted without any changes.

Reviewer #2: This is an important study investigating impact of restriction of OTC sales of antibiotics on antimicrobial resistance (AMR). This study attempts to address the most important endpoint in this type of research, but also the most challenging; Preventing increase of AMR is the main objective of regulating sales of antibiotics, but is affected by many other factors. The authors use an innovative methodology for assessing the impact of policy measures on AMR (Interrupted-time-series). However, the study still requires some work before it meets the standards of PLOS ONE. Some points of concern below.

General comments

My main concern about this study is that no data is provided about the total sales of the antibiotics that have been looked at in the study. This makes it challenging/impossible to estimate the relevance of the findings in this study. Other studies have shown that impact of regulating OTC sales of antibiotics varies greatly among countries, regions and types of antibiotics. It is essential to include the total sales volumes of cefazolin, ceftriaxone, ciprofloxacin, TMP-SMX and amikacin for the region of interest in the analysis to be able to interpret the resistance data.

A similar study has been done by Mattos et al. (Journal of Global Antimicrobial Resistance 2017). Please comment on what this study adds to the knowledge we have obtained from this study.

While the study appears to be sound, the language sometimes is unclear (for example, words like ‘straightforward’, ‘in opposite direction’ and ‘relevant’ are used incorrectly), making it difficult to follow. The whole paper needs a quick copy edit.

Title and abstract

The title is appropriate for the content of the article. The abstract is concise and accurately summarizes the essential information of the paper although it could be improved if the authors address the following issues:

• Abstract: Please include aim of the study in Introduction instead of Methods

• Abstract: Please include numbers in results for clarification of the findings. Also, the authors claim that they found an ‘impact’ on cefazolin and amikacin resistance, but this is not necessarily due to reduction of OTC sales and does not correspond with the conclusions drawn in this article.

• Abstract: Conclusion section now only exists of a policy recommendation, but no conclusion is provided based on your results.

Introduction

The introduction covers most important issues regarding banning OTC sales of antibiotics. Some points to consider:

• Unclear what you mean by ‘which have a human development index’. Please elaborate.

• No information is provided on what the actual impact of the new regulation was in the region where this study has been done.

• Further clarification is needed on why antimicrobial resistance is an important outcome, why desired impact on AMR is still uncertain and include more references on reduction of OTC sales of antibiotics in Brazil (see also comment above)

Methods

The Methods section is comprehensive and detailed but there are some points that should be clarified.

• Please explain what resistant tests have been conducted and how resistance was defined in this study.

• Why did the authors choose to analyse these specific antibiotics?

Results

It is unclear (in the first paragraph) what IRR values refer to and, more important, how to interpret these values.

Discussion

• The first paragraph reads like it is unclear whether antibiotics play a role in antimicrobial resistance at all. Moreover, the content of this paragraph seems better suited for the introduction.

• The second paragraph contains the most important text in the discussion. Would suggest to move that up to the first paragraph.

• There is a lot of focus on direct impact of banning OTC sales of antibiotics on AMR in the results sections, but this approach has some limitations since AMR resistance rates evolve and will not change from one day to another. Maybe consider mentioning this as a limitation

Conclusion

The main conclusion is very clear and well formulated. However, new information and references are quoted in the second part of the conclusion that have not been described in the discussion.

Figure

• In figure 1 for amikacin, it seems that the data is not suitable for an ITS because of the large differences between the resistance measurements. These deviating findings require clarification.

• Figure 1: Please consider including units for Resistance at Y-axis

6. PLOS authors have the option to publish the peer review history of their article (what does this mean?). If published, this will include your full peer review and any attached files.

Reviewer #1: No

Reviewer #2: **Yes: **Tom G. Jacobs

---

## [Author Response · Author response to Decision Letter 0]

15 Jun 2021

To the Editor –

We thank you and the reviewers for carefully reading our manuscript and for your useful recommendations. We responded each question bellow and changed the manuscript accordingly. 

The authors

PONE-D-21-00235

Impact of restriction of over-the-counter sales of antimicrobials on antimicrobial resistance in Escherichia coli from community-onset urinary tract infections in inner São Paulo State, Brazil.

PLOS ONE

EDITOR 

Dear Dr. Fortaleza,

Thank you for submitting your manuscript to PLOS ONE. After careful consideration, we feel that it has merit but does not fully meet PLOS ONE’s publication criteria as it currently stands. Therefore, we invite you to submit a revised version of the manuscript that addresses the points raised during the review process. Please proceed to revise the manuscript paying special attention to the comments provided by reviewer number 2. We look forward to your revised version. For Lab, Study and Registered Report Protocols: These article types are not expected to include results but may include pilot data. If you will need more time than this to complete your revisions, please reply to this message or contact the journal office at plosone@plos.org. Please include the following items when submitting your revised manuscript. A rebuttal letter that responds to each point raised by the academic editor and reviewer(s). You should upload this letter as a separate file labeled 'Response to Reviewers'.A marked-up copy of your manuscript that highlights changes made to the original version. You should upload this as a separate file labeled 'Revised Manuscript with Track Changes'. An unmarked version of your revised paper without tracked changes. You should upload this as a separate file labeled 'Manuscript'. If applicable, we recommend that you deposit your laboratory protocols in protocols.io to enhance the reproducibility of your results. Protocols.io assigns your protocol its own identifier (DOI) so that it can be cited independently in the future. For instructions see: http://journals.plos.org/plosone/s/submission-guidelines#loc-laboratory-protocols. We look forward to receiving your revised manuscript. Kind regards,

Monica Cartelle Gestal, PhD, Academic Editor. PLOS ONE. ournal Requirements: When submitting your revision, we need you to address these additional requirements. 1. Please ensure that your manuscript meets PLOS ONE's style requirements, including those for file naming. The PLOS ONE style templates can be found at

https://journals.plos.org/plosone/s/file?id=ba62/PLOSOne_formatting_sample_title_authors_affiliations.pdf 2. In ethics statement in the manuscript and in the online submission form, please provide additional information about the patient records/samples used in your retrospective study. Specifically, please ensure that you have discussed whether all data/samples were fully anonymized before you accessed them and/or whether the IRB or ethics committee waived the requirement for informed consent. If patients provided informed written consent to have data/samples from their medical records used in research, please include this information.3. Your ethics statement should only appear in the Methods section of your manuscript. If your ethics statement is written in any section besides the Methods, please move it to the Methods section and delete it from any other section. Please ensure that your ethics statement is included in your manuscript, as the ethics statement entered into the online submission form will not be publIshed alongside your manuscript. 4. Please include captions for your Supporting Information files at the end of you manuscript, and update any in-text citations to match accordingly. Please see our Supporting Information guidelines for more information: http://journals.plos.org/plosone/s/supporting-information.

AUTHORS´ RESPONSE: We changed our manuscript as required.

REVIEWERS' COMMENTS:

Reviewer's Responses to Questions

Comments to the Author 1. Is the manuscript technically sound, and do the data support the conclusions? The manuscript must describe a technically sound piece of scientific research with data that supports the conclusions. Experiments must have been conducted rigorously, with appropriate controls, replication, and sample sizes. The conclusions must be drawn appropriately based on the data presented. Reviewer #1: Yes. Reviewer #2: Partly

AUTHORS´ RESPONSE: We have changed the conclusions to be supported by the data we presented. The main conclusion is that we have a significant (p<0.001) impact after the intervention in amikacin and cefazolin. 

2. Has the statistical analysis been performed appropriately and rigorously? Reviewer #1: Yes. Reviewer #2: I Don't Know

 AUTHORS´ RESPONSE: Even though methodological choices and statistical analysis have both advantages and limitations, interrupted time series (ITS) analysis is more appropriate that usual “before and after” comparisons, since: (1) Monthly rates are not independent observations, which are assumptions of tests such as Chi-square, Student T test and Mann-Whitney U test, usually applied to “before and after analyses”; (2) ITS addresses both immediate changes and the impact on long term trends. 

3. Have the authors made all data underlying the findings in their manuscript fully available? The PLOS Data policy requires authors to make all data underlying the findings described in their manuscript fully available without restriction, with rare exception (please refer to the Data Availability Statement in the manuscript PDF file). The data should be provided as part of the manuscript or its supporting information, or deposited to a public repository. For example, in addition to summary statistics, the data points behind means, medians and variance measures should be available. If there are restrictions on publicly sharing data—e.g. participant privacy or use of data from a third party—those must be specified. Reviewer #1: Yes Reviewer #2: Yes

AUTHORS´ RESPONSE: As referred by reviewers, the database was uploaded alongside the manuscript. 

4. Is the manuscript presented in an intelligible fashion and written in standard English? PLOS ONE does not copyedit accepted manuscripts, so the language in submitted articles must be clear, correct, and unambiguous. Any typographical or grammatical errors should be corrected at revision, so please note any specific errors here. Reviewer #1: Yes. Reviewer #2: Yes

AUTHORS´ RESPONSE: We thank the reviewers.

5. Review Comments to the Author. Please use the space provided to explain your answers to the questions above. You may also include additional comments for the author, including concerns about dual publication, research ethics, or publication ethics. (Please upload your review as an attachment if it exceeds 20,000 characters) Reviewer #1: I found this manuscript very well written, in a clear and concise manner. All the findings are very well presented and the analysis of data is well done.

The findings are mostly negative which is very interesting considering how much emphasis is put on controlling antimicrobial drug use as a way to control resistance. I feel that the manuscript should be accepted without any changes. Reviewer #2: This is an important study investigating impact of restriction of OTC sales of antibiotics on antimicrobial resistance (AMR). This study attempts to address the most important endpoint in this type of research, but also the most challenging; Preventing increase of AMR is the main objective of regulating sales of antibiotics, but is affected by many other factors. The authors use an innovative methodology for assessing the impact of policy measures on AMR (Interrupted-time-series). However, the study still requires some work before it meets the standards of PLOS ONE. Some points of concern below.

General comments. My main concern about this study is that no data is provided about the total sales of the antibiotics that have been looked at in the study. This makes it challenging/impossible to estimate the relevance of the findings in this study. Other studies have shown that impact of regulating OTC sales of antibiotics varies greatly among countries, regions and types of antibiotics. It is essential to include the total sales volumes of cefazolin, ceftriaxone, ciprofloxacin, TMP-SMX and amikacin for the region of interest in the analysis to be able to interpret the resistance data. A similar study has been done by Mattos et al. (Journal of Global Antimicrobial Resistance 2017). Please comment on what this study adds to the knowledge we have obtained from this study.

AUTHORS´ RESPONSE: We thank the reviewers. Unfortunately, we had no access to antimicrobial sales database (which are not currently available from governmental databases.). The present study was based on microbiology results that are representative of overall community-onset urinary tract infection in this area. We acknowledge that the study by Mattos et al, ‘‘Brazil’s resolutions to regulate the sale of antibiotics: Impact on consumption and Escherichia coli resistance rates’’ also analyzed the impact of RDC No. 44/2010 on the rates of antimicrobial resistance. Their results showed that antimicrobial resistance rates increased for almost all classes of antibiotics tested and that antimicrobial resistance did not appear to be influenced by the evaluated resolutions (RDCs 44/2010 and 20/2011). Their study includes data from 2009 to 2015, with intervention instituted in October 2010. From this, we emphasize that our study is important because it includes a larger time interval, since our data is from 2005 to 2018. The pre-intervention period on our study, in comparison to theirs, is extended, which adds knowledge to Mattos et al study. Considering that changes in antimicrobial resistance evolve, we believe this larger period analysis is an important advantage of our study. In addition to that, our study differs from his because ours analyzes the impact of RDCs no. 44/2010 in the state of São Paulo, region of (Botucatu). It is a different region (Campinas) from the one evaluated by the study of Mattos et al. Mattos et al found that antimicrobial resistance does not seem to be affected by RDCs, a similar result to ours, since we have also concluded that this resolution by itself was not sufficient to cause a statistically significant impact. In our conclusion, we emphasize the importance of evaluating some additional geographic data, including the use of antimicrobials in agriculture and the different populational patterns of gut microbiota. Thus, we conclude that analyzing a population residing in another location in the state of São Paulo is also a positive point of our study.In the study by Mattos et. al, the sale of antimicrobials from 2008 to 2012 was evaluated, and the samples analyzed varied from 2009 to 2015. Despite the disparity between the analysis intervals of these two aspects, we consider that there was no prejudice to the final conclusion of the study. At the same time, we understand the usefulness of assessing the impact of the resolution on the sale of antibiotics and included the lack of sales data as a limitatipn of our study.The desired impact on antimicrobial resistance is still uncertain in Brazil because there is a lack of regional studies that analyze the rates of antimicrobial resistance throughout the country, using RDC 44/2010 as an intervention. Brazil is the fifth largest country in the world and is characterized by important social inequality, with very large regional disparities. In this way, our country has areas with very different demographic, social and economic characteristics, and each region of the country, in fact, presents a productive peculiarity. Considering that, we believe only legislation changes on the sale of antibiotics was not sufficient as an isolated method to fight against increase of antimicrobial resistance, which is reinforced by the study by Mattos et al.

While the study appears to be sound, the language sometimes is unclear (for example, words like ‘straightforward’, ‘in opposite direction’ and ‘relevant’ are used incorrectly), making it difficult to follow. The whole paper needs a quick copy edit.

AUTHORS´ RESPONSE: a quick edit was done by the authors.

Title and abstract. The title is appropriate for the content of the article. The abstract is concise and accurately summarizes the essential information of the paper although it could be improved if the authors address the following issues:• Abstract: Please include aim of the study in Introduction instead of Methods • Abstract: Please include numbers in results for clarification of the findings. Also, the authors claim that they found an ‘impact’ on cefazolin and amikacin resistance, but this is not necessarily due to reduction of OTC sales and does not correspond with the conclusions drawn in this article.• Abstract: Conclusion section now only exists of a policy recommendation, but no conclusion is provided based on your results.

AUTHORS´ RESPONSE: We agree with recommendations and changed the text accordingly.

Introduction. The introduction covers most important issues regarding banning OTC sales of antibiotics. Some points to consider: • Unclear what you mean by ‘which have a human development index’. Please elaborate. • No information is provided on what the actual impact of the new regulation was in the region where this study has been done. • Further clarification is needed on why antimicrobial resistance is an important outcome, why desired impact on AMR is still uncertain and include more references on reduction of OTC sales of antibiotics in Brazil (see also comment above)

AUTHORS´ RESPONSE: We agree with recommendations and changed the text accordingly.

Methods. The Methods section is comprehensive and detailed but there are some points that should be clarified.• Please explain what resistant tests have been conducted and how resistance was defined in this study.• Why did the authors choose to analyse these specific antibiotics?

AUTHORS´ RESPONSE: We included the information as requested.

Results It is unclear (in the first paragraph) what IRR values refer to and, more important, how to interpret these values.

AUTHORS´ RESPONSE: We expanded the result sections to explain and provide interpretations for those findings. 

Discussion • The first paragraph reads like it is unclear whether antibiotics play a role in antimicrobial resistance at all. Moreover, the content of this paragraph seems better suited for the introduction. • The second paragraph contains the most important text in the discussion. Would suggest to move that up to the first paragraph. • There is a lot of focus on direct impact of banning OTC sales of antibiotics on AMR in the results sections, but this approach has some limitations since AMR resistance rates evolve and will not change from one day to another. Maybe consider mentioning this as a limitation. Conclusion. The main conclusion is very clear and well formulated. However, new information and references are quoted in the second part of the conclusion that have not been described in the discussion.

AUTHORS´ RESPONSE: We expanded the limitation section and rewrote the second part of the conclusion. 

Figure • In figure 1 for amikacin, it seems that the data is not suitable for an ITS because of the large differences between the resistance measurements. These deviating findings require clarification. • Figure 1: Please consider including units for Resistance at Y-axis

AUTHORS´ RESPONSE: We understand that the variation limits the ITS analysis of Amikacin. We chose to include it as a limitation in the footnote.

6. PLOS authors have the option to publish the peer review history of their article (what does this mean?). If published, this will include your full peer review and any attached files. If you choose “no”, your identity will remain anonymous, but your review may still be made public. Do you want your identity to be public for this peer review? For information about this choice, including consent withdrawal, please see our Privacy Policy. Reviewer #1: No Reviewer #2: Yes: Tom G. Jacobs

AUTHORS´ RESPONSE: We thank once again the reviewers for their recommendations.

---

## [Decision Letter · Decision Letter 1]

24 Aug 2021

PONE-D-21-00235R1

Impact of restriction of over-the-counter sales of antimicrobials on antimicrobial resistance in Escherichia coli from community-onset urinary tract infections in inner São Paulo State, Brazil.

PLOS ONE

Dear Dr. Fortaleza,

Thank you for submitting your manuscript to PLOS ONE. After careful consideration, we feel that it has merit but does not fully meet PLOS ONE’s publication criteria as it currently stands. Therefore, we invite you to submit a revised version of the manuscript that addresses the points raised during the review process.

Thanks for your submission, please address the reviewers comments and submit as early as your convenience

We look forward to receiving your revised manuscript.

Kind regards,

Monica Cartelle Gestal, PhD

Academic Editor

PLOS ONE

Journal Requirements:

Reviewers' comments:

Reviewer's Responses to Questions

**Comments to the Author**

1. If the authors have adequately addressed your comments raised in a previous round of review and you feel that this manuscript is now acceptable for publication, you may indicate that here to bypass the “Comments to the Author” section, enter your conflict of interest statement in the “Confidential to Editor” section, and submit your "Accept" recommendation.

Reviewer #2: All comments have been addressed

Reviewer #3: All comments have been addressed

2. Is the manuscript technically sound, and do the data support the conclusions?

Reviewer #2: Yes

Reviewer #3: Yes

3. Has the statistical analysis been performed appropriately and rigorously? 

Reviewer #2: I Don't Know

Reviewer #3: I Don't Know

4. Have the authors made all data underlying the findings in their manuscript fully available?

Reviewer #2: Yes

Reviewer #3: Yes

5. Is the manuscript presented in an intelligible fashion and written in standard English?

Reviewer #2: Yes

Reviewer #3: No

6. Review Comments to the Author

Reviewer #2: I am happy with the adjustments made by the authors and their responses to the questions. I support publication of this paper in PLOS ONE.

Although, in general, the level of scientific English in the manuscript is appropriate, there are still a few typo's and grammatical errors.

Reviewer #3: Dear Authors the following statements are ambiguous

1. on the conclusion section of the abstract ''At the present study, we found that OTC sales restriction did not

generally impact on antimicrobial resistance. '' = it is better rewritten in a more informative way.

2.the last statement of the first paragraph on the background section ''Laws restricting

or prohibiting OTC sales of antimicrobials have been issued, but their impact is

currently not straightforward. (it is better to paraphrase the phrase -----currently not straightforward)

3.on the first statement of 2nd paragraph of the background section---''The population drivers of antimicrobial resistance are complex''. (it is better to change the phrase population drivers to community drivers)

4.the last statement of the third paragraph of the background section ''Most importantly, the impact on antimicrobial

resistance is uncertain in Brazil and in our region.(better to say---the impact on antimicrobial resistance is uncertain in Brazil generally and in our region specifically)

5. the statement on the discussion section ''Disappointing as our findings may be, they are evidently not an argument against

restricting OTC sales of antimicrobials.'' is ambiguous. I advise the authors to make it more clear.

7. PLOS authors have the option to publish the peer review history of their article (what does this mean?). If published, this will include your full peer review and any attached files.

Reviewer #2: **Yes: **Tom G Jacobs

Reviewer #3: **Yes: **Oumer Abdu Muhie

---

## [Author Response · Author response to Decision Letter 1]

30 Aug 2021

1. REVIEWERS' COMMENTS: If the authors have adequately addressed your comments raised in a previous round of review and you feel that this manuscript is now acceptable for publication, you may indicate that here to bypass the “Comments to the Author” section, enter your conflict of interest statement in the “Confidential to Editor” section, and submit your "Accept" recommendation. 

Reviewer #2: All comments have been addressed 

Reviewer #3: All comments have been addressed 

AUTHORS´ RESPONSE: We thank the reviewers

2. REVIEWERS' COMMENTS: Is the manuscript technically sound, and do the data support the conclusions? 

Reviewer #2: Yes 

Reviewer #3: Yes 

AUTHORS´ RESPONSE: We thank the reviewers

3. REVIEWERS' COMMENTS: Has the statistical analysis been performed appropriately and rigorously? 

Reviewer #2: I Don't Know 

Reviewer #3: I Don't Know 

AUTHORS´ RESPONSE: a natural experiment based on ecological data was the chosen design. The statistical analysis for interrupted time series (ITS) was performed in STATA 14 (Statacorp, College Station, TX). Seasonality was investigated by Box-Jenkins models in NCSS 9. 

4. REVIEWERS' COMMENTS: Have the authors made all data underlying the findings in their manuscript fully available? 

Reviewer #2: Yes 

Reviewer #3: Yes 

AUTHORS´ RESPONSE: We thank the reviewers

5. REVIEWERS' COMMENTS: Is the manuscript presented in an intelligible fashion and written in standard English? 

Reviewer #2: Yes 

Reviewer #3: No 

AUTHORS´ RESPONSE: the authors have reviewed all the text and corrected some typo and grammatical errors. 

6. REVIEWERS' COMMENTS: Review Comments to the Author 

Reviewer #2: I am happy with the adjustments made by the authors and their responses to the questions. I support publication of this paper in PLOS ONE.

Although, in general, the level of scientific English in the manuscript is appropriate, there are still a few typos and grammatical errors. 

AUTHORS´ RESPONSE: We thank the reviewer

Reviewer #3: Dear Authors the following statements are ambiguous:

1. on the conclusion section of the abstract ''At the present study, we found that OTC sales restriction did not generally impact on antimicrobial resistance. '' = it is better rewritten in a more informative way.

AUTHORS´ RESPONSE: we agree with recommendations and changed the manuscript according to suggestion.

2.the last statement of the first paragraph on the background section ''Laws restricting or prohibiting OTC sales of antimicrobials have been issued, but their impact is currently not straightforward. (it is better to paraphrase the phrase -----currently not straightforward)

AUTHORS´ RESPONSE: we agree with recommendations and changed the manuscript according to suggestion.

3.on the first statement of 2nd paragraph of the background section---''The population drivers of antimicrobial resistance are complex''. (it is better to change the phrase population drivers to community drivers)

AUTHORS´ RESPONSE: we agree with recommendations and changed the manuscript according to suggestion.

4.the last statement of the third paragraph of the background section ''Most importantly, the impact on antimicrobial resistance is uncertain in Brazil and in our region. (better to say---the impact on antimicrobial resistance is uncertain in Brazil generally and in our region specifically)

AUTHORS´ RESPONSE: we agree with recommendations and changed the manuscript according to suggestion.

5. the statement on the discussion section ''Disappointing as our findings may be, they are evidently not an argument against

restricting OTC sales of antimicrobials.'' is ambiguous. I advise the authors to make it more clear. 

AUTHORS´ RESPONSE: we changed the manuscript according to suggestion. Our study did not found evidence of the impact of OTC sales restriction and antimicrobial resistance, but, we don’t have any arguments against the restriction.

---

## [Decision Letter · Decision Letter 2]

25 Oct 2021

Impact of restriction of over-the-counter sales of antimicrobials on antimicrobial resistance in Escherichia coli from community-onset urinary tract infections in inner São Paulo State, Brazil.

PONE-D-21-00235R2

Dear Dr. Fortaleza,

We’re pleased to inform you that your manuscript has been judged scientifically suitable for publication and will be formally accepted for publication once it meets all outstanding technical requirements.

Kind regards,

Monica Cartelle Gestal, PhD

Academic Editor

PLOS ONE

Additional Editor Comments (optional):

Reviewers' comments:

Reviewer's Responses to Questions

**Comments to the Author**

1. If the authors have adequately addressed your comments raised in a previous round of review and you feel that this manuscript is now acceptable for publication, you may indicate that here to bypass the “Comments to the Author” section, enter your conflict of interest statement in the “Confidential to Editor” section, and submit your "Accept" recommendation.

Reviewer #2: All comments have been addressed

Reviewer #3: All comments have been addressed

2. Is the manuscript technically sound, and do the data support the conclusions?

Reviewer #2: Yes

Reviewer #3: Yes

3. Has the statistical analysis been performed appropriately and rigorously? 

Reviewer #2: I Don't Know

Reviewer #3: I Don't Know

4. Have the authors made all data underlying the findings in their manuscript fully available?

Reviewer #2: Yes

Reviewer #3: Yes

5. Is the manuscript presented in an intelligible fashion and written in standard English?

Reviewer #2: Yes

Reviewer #3: Yes

6. Review Comments to the Author

Reviewer #2: Happy with the changes made by the authors. THe manuscript has considerably been improved since their initial submission.

Reviewer #3: (No Response)

7. PLOS authors have the option to publish the peer review history of their article (what does this mean?). If published, this will include your full peer review and any attached files.

Reviewer #2: **Yes: **Tom G Jacobs

Reviewer #3: **Yes: **Oumer Abdu Muhie

---

## [Editor Report · Acceptance letter]

29 Oct 2021

PONE-D-21-00235R2 

Impact of restriction of over-the-counter sales of antimicrobials on antimicrobial resistance in *Escherichia coli* from community-onset urinary tract infections in inner São Paulo State, Brazil. 

Dear Dr. Fortaleza:

I'm pleased to inform you that your manuscript has been deemed suitable for publication in PLOS ONE. Congratulations! Your manuscript is now with our production department. 

Kind regards, 

on behalf of

Dr. Monica Cartelle Gestal 

Academic Editor

PLOS ONE